# Reductive Activity and Mechanism of Hypoxia- Targeted AGT Inhibitors: An Experimental and Theoretical Investigation

**DOI:** 10.3390/ijms20246308

**Published:** 2019-12-13

**Authors:** Weinan Xiao, Guohui Sun, Tengjiao Fan, Junjun Liu, Na Zhang, Lijiao Zhao, Rugang Zhong

**Affiliations:** 1Beijing Key Laboratory of Environmental & Viral Oncology, College of Life Science and Bioengineering, Beijing University of Technology, Beijing 100124, China; xwn@emails.bjut.edu.cn (W.X.); sunguohui@bjut.edu.cn (G.S.); fantengjiao2014@emails.bjut.edu.cn (T.F.); liujunjun@emails.bjut.edu.cn (J.L.); nanatonglei@bjut.edu.cn (N.Z.); lifesci@bjut.edu.cn (R.Z.); 2Department of Medical Technology, Beijing Pharmaceutical University of Staff and Workers, Beijing 100079, China

**Keywords:** hypoxia-activated AGT inhibitors, reduction of nitro group, density functional theory, HPLC-ESI-MS/MS, molecular docking, tumor targeting

## Abstract

O^6^-alkylguanine-DNA alkyltransferase (AGT) is the main cause of tumor cell resistance to DNA-alkylating agents, so it is valuable to design tumor-targeted AGT inhibitors with hypoxia activation. Based on the existing benchmark inhibitor O^6^-benzylguanine (O^6^-BG), four derivatives with hypoxia-reduced potential and their corresponding reduction products were synthesized. A reductase system consisting of glucose/glucose oxidase, xanthine/xanthine oxidase, and catalase were constructed, and the reduction products of the hypoxia-activated prodrugs under normoxic and hypoxic conditions were determined by high-performance liquid chromatography electrospray ionization tandem mass spectrometry (HPLC-ESI-MS/MS). The results showed that the reduction products produced under hypoxic conditions were significantly higher than that under normoxic condition. The amount of the reduction product yielded from ANBP (2-nitro-6-(3-amino) benzyloxypurine) under hypoxic conditions was the highest, followed by AMNBP (2-nitro-6-(3-aminomethyl)benzyloxypurine), 2-NBP (2-nitro-6-benzyloxypurine), and 3-NBG (O6-(3-nitro)benzylguanine). It should be noted that although the levels of the reduction products of 2-NBP and 3-NBG were lower than those of ANBP and AMNBP, their maximal hypoxic/normoxic ratios were higher than those of the other two prodrugs. Meanwhile, we also investigated the single electron reduction mechanism of the hypoxia-activated prodrugs using density functional theory (DFT) calculations. As a result, the reduction of the nitro group to the nitroso was proven to be a rate-limiting step. Moreover, the 2-nitro group of purine ring was more ready to be reduced than the 3-nitro group of benzyl. The energy barriers of the rate-limiting steps were 34–37 kcal/mol. The interactions between these prodrugs and nitroreductase were explored via molecular docking study, and ANBP was observed to have the highest affinity to nitroreductase, followed by AMNBP, 2-NBP, and 3-NBG. Interestingly, the theoretical results were generally in a good agreement with the experimental results. Finally, molecular docking and molecular dynamics simulations were performed to predict the AGT-inhibitory activity of the four prodrugs and their reduction products. In summary, simultaneous consideration of reduction potential and hypoxic selectivity is necessary to ensure that such prodrugs have good hypoxic tumor targeting. This study provides insights into the hypoxia-activated mechanism of nitro-substituted prodrugs as AGT inhibitors, which may contribute to reasonable design and development of novel tumor-targeted AGT inhibitors.

## 1. Introduction

DNA-alkylating agents such as temozolomide (TMZ) and carmustine (BCNU) are an important class of anticancer drugs for the treatment of malignant tumors [1,2,3,4]. They exert anticancer effects by causing DNA alkylation, blocking the normal replication or transcription of DNA, and inducing apoptosis of tumor cells [5,6,7,8,9]. However, due to the presence of intracellular O^6^-alkylguanine-DNA alkyltransferase (AGT), which can repair the alkylation damage caused by such agents, the formation of mono-adducts or lethal dG-dC inter-crosslinks can be blocked, leading to drug resistance [10,11,12,13,14,15]. Therefore, inhibiting AGT activity in tumor cells is of great significance for improving the chemotherapeutic effects of treatment. Over the past few decades, a series of AGT inhibitors have been synthesized for combination chemotherapy with alkylating agents [3,16,17,18,19,20,21,22,23]. Among them, O^6^-benzylguanine (O^6^-BG) was the first AGT inhibitor to enter clinical trials with superior AGT inhibitory activity [3,15,24]. Although O^6^-BG opened up a new field for enhanced tumor chemotherapy, clinical studies showed that it also inhibits the AGT activity in normal cells, which in turn leads to a significant increase in the myelosuppressive effects and consequently reduces the chemotherapeutic effects of such drugs [25]. Therefore, the development of tumor-targeted AGT inhibitors is promising for improving the anticancer effects of these alkylating agents.

The rapid proliferation of tumor cells leads to high oxygen consumption and the structural dysfunction of tumor vasculature causes insufficient oxygen supply, making intratumoral hypoxia one of the important features of solid tumors [26,27,28,29,30]. Therefore, the design of bioreduction prodrugs based on the microenvironment of tumor hypoxia has become a new strategy to improve the efficacy of chemotherapy [31]. At present, some researchers have developed a series of hypoxia-activated prodrugs showing inhibitory effects on tumor cells in vivo or in vitro [32,33,34,35,36,37,38,39,40,41,42,43,44]. These prodrugs are not or less toxic to cells under normoxic condition, while under hypoxic conditions, they can be reduced by reductases in tumor cells to form cytotoxic agents. In contrast, they will be re-oxidized to form the prodrugs itself under normoxic condition [32]. Generally, hypoxia-activated prodrugs include nitro compounds, N-oxides, quinones, and metal complexes [33]. Nitro prodrugs can be efficiently reduced to nitroso, hydroxylamine (four electrons), and amino intermediates or products (six electrons) by a series of single-electron reduction pathways under hypoxic conditions [34,35,36,37,38,39,40]. Based on this feature, various nitroaromatic and nitroheterocyclic bioreduction prodrugs can be obtained. Sartorelli’s group [41,42,43,44] designed and synthesized a class of hypoxia-activated AGT inhibitor prodrugs, which were inactive against AGT or showed very low activity under normoxic conditions, but under hypoxic condition can be activated by reductase and release an effective AGT inhibitor.

In this study, we synthesized four hypoxia-activated O^6^-BG derivatives as prodrugs and their reduction products (Figure 1), including O^6^-(3-nitro)benzylguanine (3-NBG), 2-nitro-6-benzyloxypurine (2-NBP), 2-nitro-6-(3-amino)benzyloxypurine (ANBP), and 2-nitro-6-(3-aminomethyl)benzyloxypurine (AMNBP), and two corresponding reduction products, O^6^-(3-amino)benzylguanine (ABG) and O^6^-(3-aminomethyl)benzylguanine (AMNBG). O^6^-BG as the reduction product of 2-NBP was purchased from a commercial supplier. A reductase system was then constructed and the reduction products of these four prodrugs were quantified using high performance liquid chromatography–electrospray ionization tandem mass spectrometry (HPLC-ESI-MS/MS) under normoxia and hypoxia. In a theoretical study, the single electron reduction mechanism of the prodrugs was investigated using quantum chemistry calculations. Furthermore, the interactions between the prodrugs and nitroreductase or AGT were analyzed by molecular docking and molecular dynamics (MD) simulations.

## 2. Results and Discussion

### 2.1. Reduction of 3-NBG, 2-NBP, ANBP, and AMNBP under Hypoxic and Normoxic Conditions

Four prodrugs (3-NBG, 2-NBP, ANBP, and AMNBP) and two reduction products (ABG and AMBG) were synthesized (Appendix A) in our laboratory, and O^6^-BG as the reduction product of 2-NBP was purchased from J & K Chemical. An in vitro reductase system was constructed under normoxic and hypoxic conditions and the reactions were terminated at 0.25, 0.5, 0.75, 1, 1.5, 2, 2.5, and 3 h. Figure 2 shows the typical selected reaction monitoring (SRM) chromatograms of ABG, AMBG, and O^6^-BG yielded from the prodrugs and the internal standard D_6_-O^6^-BG. All samples were divided into control groups (Figure 2A,D,G), reduction groups under normoxic conditions (Figure 2B,E,H), and reduction groups under hypoxic condition (Figure 2C,F,I). The fractions of ABG, AMBG, and O^6^-BG were eluted at 13, 12, and 18 min, respectively, accompanied by D_6_-O^6^-BG eluted at 18 min. No signal for the fractions of ABG, AMBG, and O^6^-BG was observed from any control sample, which indicated that there was no significant matrix interference or contamination in the analyte channels from the reductase system or the internal standard. Linear correlation coefficients higher than 0.999 were obtained for the quantitation of the reduction products in view of calibration curves (Appendix A).

As illustrated in Figure 3, at the same concentrations of the prodrugs, the reduction products under hypoxic conditions were observed with higher levels than those under normoxic conditions (data listed in Appendix A). It was obvious that the reduction potential of the four prodrugs under hypoxic condition was higher than it was under normoxic conditions. Under hypoxic conditions, the amount of reduction products yielded followed the order of ANBP > AMNBP > 2-NBP > 3-NBG. Through the high concentrations of ABG (reduction from ANBP, Figure 3C) and AMBG (reduction from AMNBP, Figure 3D), ANBP and AMNBP exhibited strong reduction potential and certain drug dose-dependence but limited superiority under hypoxic conditions over normoxic conditions. Unlike ANBP and AMNBP, 3-NBG and 2-NBP showed low reductive activity, but their reductive activity under hypoxic conditions was considerably superior to that under normoxic conditions. As shown in Figure 3A,B, the yield of the reduction products at low concentration (5 mM) under hypoxic conditions was significantly higher (*p* < 0.01) than that of high-concentration prodrugs (10 mM) under normoxic conditions. In addition, the maximum ratios of hypoxia to normoxia of 3-NBG and 2-NBP were higher (3-NBG: C_hypoxic_/C_normoxic_ = 5.55 at 3 h, 2-NBP: C_hypoxic_/C_normoxic_ = 6.01 at 1.5 h) than those of ANBP and AMNBP, suggesting that 3-NBG and 2-NBP had better hypoxia selectivity. In summary, distinct reduction potential and hypoxia selectivity were observed in the four prodrugs, which were all O^6^-BG derivatives containing a common nitro group at different position. We speculated that this difference might have been related to the chemical structure, reaction energy, and interaction between the molecules involved in the reduction mechanism of the prodrugs. Consequently, quantum chemistry calculations and molecular docking were carried out to try to explain the experimental phenomena.

### 2.2. Quantum Chemistry Calculations

#### 2.2.1. Mechanism of Single-Electron Reduction Using Nitrobenzene as A Model Compound

In this study, we selected nitrobenzene as a simplified model compound to investigate the single-electron reduction mechanism of the present hypoxia-activated prodrugs containing a nitro moiety as the triggering group. The whole reaction of nitrobenzene to aniline requires in total six electrons and six protons (Figure 4), which can be divided into three steps and six transition states (TSs) may be involved. In the first step, the nitro group was reduced to nitroso intermediate (IC_2_) by transferring 2e^−^/2H^+^, and a water molecule was eliminated. In the second step, with the transfer of another 2e^−^/2H^+^, IC_2_ was converted to a hydroxylamine intermediate (IC_4_). Finally, IC_4_ received the last 2e^−^/2H^+^, followed by the production of aniline while eliminating a water molecule [45,46,47,48,49,50,51]. Generally, the reduction of nitrobenzene is mediated by nitroreductase, where reduced flavin mononucleotide (FMNH) is located at the active center as a coenzyme. Considering computational complexity, the molecular structure of FMNH was simplified by replacing the phosphate tail chain on the *iso*-oxazine ring with a methyl group.

For the structures of the transition states involved in the whole reaction (Appendix A), the N–H bond length at the N5 position of FMNH was relatively increased compared to the original bond length (1.01 Å), indicating that the H at the N5 position of FMNH had a tendency to leave. In each step, the distance between the O of the nitro group and the H at the N5 position of FMNH decreased, and then the N–O bond length increased. Finally, two H atoms were added, followed by the formation of the amino group. As shown in Figure 5 (data listed in Appendix A), the relative energy of the reaction showed the same trend at the four theoretical levels. Except the step of nitrobenzene hydroxylized to IC_1_, which was endothermic, the steps were all exothermic. The dehydration of IC_1_ to produce nitrosobenzene required a maximum energy barrier (36.57 kcal/mol at the CPCM-M062X/6-311+G(3df,2p) theoretical level), so the step overcoming TS_2_ was considered a rate-limiting step, which was consistent with previously reported studies [46,52]. Interestingly, TS_5_ also had a high energy barrier (36.33 kcal/mol at the CPCM-M062X/6-311+G(3df,2p) theoretical level). According to a previous report [51], the presence of a hydroxylamine intermediate (IC_4_) was detected in the reduction of the nitro group. Therefore, the reaction overcoming TS_5_ is the rate-limiting step in the reaction of hydroxylamine to aniline.

#### 2.2.2. Reduction Mechanisms of 3-NBG, 2-NBP, ANBP, and AMNBP

Similarly to nitrobenzene, the reduction of 3-NBG mediated by coenzyme FMNH also involves the transfer of 6e^−^/6H^+^ and breaking of two N–O bonds. The OH• radicals produced were reduced to water as byproducts (Figure 6). Briefly, the step of hydroxylation to form IC_1_ was an endothermic reaction and the step of dehydration producing nitroso intermediates needed to overcome the highest energy barrier (36.88 kcal/mol at the CPCM-M062X/6-311+G(3df,2p) theoretical level), so TS_2_ was the rate-limiting step in the entire reaction, as for nitrobenzene (Figure 7) (data listed in Appendix A). Based on the similar reduction processes of 3-NBG and nitrobenzene, the reduction mechanisms of the other three prodrugs (2-NBP, ANBP, and AMNBP) were explored by only calculating their first two steps (from RC to IC_2_), including the rate-limiting step.

As shown in Figure 8, after transferring 2e^−^/2H^+^, 2-NBP, ANBP, and AMNBP were reduced to the corresponding nitroso intermediates by FMNH coenzyme. The relative energy of the three prodrugs were little different (less than 1 kcal/mol) from each other (Figure 9) (data listed in Appendix A). Table 1 lists the energy barrier of the rate-limiting step (TS_2_) for the four prodrugs (3-NBG, 2-NBP, ANBP, and AMNBP) calculated at different theoretical levels. In general, the energy barrier of 3-NBG was ~2–5 kcal/mol higher than those of the other three prodrugs at all theoretical levels, which indicated that the 2-nitro group of the purine ring was more reactive than the 3-nitro group of the benzyl. This result was consistent with the phenomenon observed in the reduction assay that 3-NBG exhibited the lowest reductive activity and yielded the smallest amount of reduction product. Although the theoretical results suggested that a different substituted position of the nitro group could lead to a different reduction potential, the order of energy barriers of the rate-limiting steps was insufficient to explain the reductive activity (ANBP > AMNBP > 2-NBP > 3-NBG) observed in the above experiments. In order to further clarify the experimental results, the interactions between the four prodrugs and nitroreductase were explored by molecular docking.

### 2.3. Docking Results

Molecular docking studies were performed to understand the action modes of the four prodrugs on nitroreductase by employing the X-ray crystal structure of the nitroreductase protein containing FMNH coenzyme (PDB entry: 4PU0) as a model. The binding affinities of the four prodrugs with the receptor are presented by the docking scores, and the hydrogen bonds between the prodrugs and the active center of the enzyme are listed in Table 2. The docked conformations of the four prodrugs with nitroreductase protein are shown in Figure 10.

The four prodrugs presented similar conformations, i.e., the guanine moiety of the four prodrugs paralleled the isoalloxazine ring of FMNH and two hydrogen bonds were observed between Glu268 and each prodrug. For the four prodrugs, 3-NBG and 2-NBP formed only two hydrogen bonds with Glu268 and had the lowest scores (Table 2), while one additional hydrogen bond was formed between AMNBP and Leu266. For the docked complex of ANBP with nitroreductase, five hydrogen bonds in total were observed within the active pocket and the highest Gold score was obtained. Besides the three hydrogen bonds formed with Glu268 and Leu266 in the AMNBP–nitroreductase complex, two new hydrogen bonds were observed on the Lys78 and Asp264 residuals in the ANBP–nitroreductase complex. The results indicated that ANBP showed the best affinity with nitroreductase, followed by AMNBP, 2-NBP, and 3-NBG, which was in a good agreemt with the reductive activity observed in the in vitro experiments. Although the difference between the docking results of 2-NBP and 3-NBG with nitroreductase was slight, the energy barrier for the reduction of 3-NBG was obviously higher than that of 2-NBP (Table 1). Consequently, the results of the docking analysis and DFT calculations together supported the reductive activity observed in the experiments as ANBP > AMNBP > 2-NBP > 3-NBG.

To predict the AGT-inhibitory capability, docking was also performed for the four prodrugs and their reduction products binding to AGT protein. The crystal structure of AGT was obtained from the Protein Database Bank with the PDB entry 1QNT. As shown in Figure 11A–D, the four prodrugs were docked into the active pocket of AGT, but they displayed an incorrect orientation compared to the pose of the ligand in the crystal structure of the AGT–DNA complex containing O^6^-methylguanine (PDB entry: 1T38), and none of them formed effective hydrogen bonds with the AGT active center (Appendix A). This suggested that the prodrugs were inactive against AGT under normoxic conditions, since only the reductive form inhibits AGT activity. In contrast, the three reduction products, ABG, AMBG, and O^6^-BG, docked reasonably with AGT, and the structures of the obtained complexes are shown in Figure 11E–G. The reduction products took the correct orientation as the crystal structure of AGT–DNA complex with PDB entry 1T38 (Appendix A), and several hydrogen bonds were formed between the ligands and the amino acid residuals in the active pocket (Table 3). Four hydrogen bonds were formed between O^6^-BG and Tyr114, Cys145, Val148, and Ser159, which was consistent with the results of previous studies [52,53,54]. Five hydrogen bonds were observed between ABG or AMBG and AGT. A hydrogen bond was not formed between ABG and Tyr114, but two new hydrogen bonds were formed at ABG·Lys165 and ABG·Arg135. Besides the four residues Tyr114, Cys145, Val148, and Ser159, a new hydrogen bond was observed between AMBG and Asn137. In addition, the 2D model of binding interactions between O^6^-BG or ABG or AMBG and AGT also produced similar results (Appendix A). All three reduction products could interact with Cys145 via one Pi–sulfur interaction. Four Pi–alkyl interactions were observed between O^6^-BG or ABG or AMBG and the amino acid residues in the active pocket of AGT. Although O^6^-BG could interact with Ser159 via one more Pi–lone pair, ABG and AMBG formed more hydrogen bonds. Therefore, ABG and AMBG might possess increased affinity with AGT due to the formation of an additional hydrogen bond compared with O^6^-BG. Based on the docking results, it can be speculated that the four prodrugs may be able to exert AGT-inhibitory activity in the hypoxic microenvironment of solid tumors, and consequently may have applications in combination chemotherapy to inhibit AGT-mediated drug resistance.

### 2.4. Molecular Dynamics Simulation and Free Energy Calculation

To further investigate the activity of ABG, AMBG, and O^6^-BG as AGT inhibitors, MD simulation and the relative binding–free energy calculations were performed using the Amber 14 program package. From 60 ns conventional MD simulations in explicit water, we checked the stability of three AGT–ligand (ABG, AMBG, and O^6^-BG) systems. Figure 12 demonstrates the time-dependent root mean square deviation (RMSD) profile of each trajectory from the minimized structure. Although all AGT–ligand systems were observed to be stable during the last 20 ns, the fluctuations of the RMSD values of AGT in all systems showed a different trend throughout the simulations. It was obvious that the RMSD values of O^6^-BG suddenly increased from 0.5 to 1.24 Å in 15 ns. In contrast, those of ABG and AMBG maintained stable fluctuations. In the molecular docking result, the amino group on the benzyl of ABG and AMBG formed hydrogen bonds with Tyr114 and Asn137, respectively, but O^6^-BG did not. Thus, it was speculated that the hydrogen bond between amino group and residue promoted the stability of ABG and AMBG. The average RMSD values of ABG (0.3 Å) and AMBG (0.5 Å) systems were lower than that of O^6^-BG (1.24 Å), which also suggested that ABG and AMBG were more stable than O^6^-BG in the AGT active pocket. The relative binding free energies obtained from the molecular mechanics/Poisson–Boltzmann surface area (MM/PBSA) calculations for the three AGT–ligand systems are listed in Table 4. The total ΔG of O^6^-BG binding with AGT was −32.96 kcal/mol, which was 2.67 and 2.49 kcal/mol higher than those of ABG and AMBG, respectively. Interestingly, the MD simulations and the free energy calculations were consistent with molecular docking results, which both indicated that ABG and AMBG have superior AGT protein affinity, structural stability, and free binding energy compared to O^6^-BG. Thus, ABG and AMBG have higher activity than O^6^-BG as AGT inhibitors.

## 3. Materials and Methods

### 3.1. Experimental Study

#### 3.1.1. Chemicals and Reagents

All chemicals, solvents and reagents were purchased from J & K Scientific Ltd. (Beijing, China), Sigma Aldrich Trading Co., Ltd. (Shanghai, China) and Beijing Chemicals Co. (Beijing, China), and were used without further purification. D_6_-O^6^-BG was synthesized in our laboratory [55].

^1^H NMR spectra were recorded on a BRUKER AC-P400 spectrometer (400 MHz) with tetramethylsilane as an internal standard. High-resolution mass spectra were recorded on a maXis QTOF high-resolution mass spectrometer (Bruker Daltonics, Bremen, Germany). IR spectra were recorded on a BRUKER VERTEX 70 Fourier-transform infrared spectrometer (Bruker Daltonics, Bremen, Germany). UV spectra were recorded on a TU-1901 Dual Beam UV-Vis Spectrophotometer from Beijing Purkinje General Instrument Co., Ltd. (Beijing, China).

#### 3.1.2. Chemical Syntheses

Six derivatives of O^6^-BG including 2-NBP (Appendix A), ANBP (Appendix A), AMNBP (Appendix A), 3-NBG (Appendix A), ABG (Appendix A), and AMBG (Appendix A) were synthesized, of which 2-NBP and AMBG have been reported previously [21,43]. The detailed synthesis information is described in the Appendix A.

#### 3.1.3. Reduction of the Four Prodrugs in Normoxic or Hypoxic Conditions

To a solution of PBS (425 μL, 100 mM, pH 7.4) was added glucose (50 μL, 100 mM), glucose oxidase (5 μL, 1000 U/mL), catalase (5 μL, 60,000 U/mL), xanthine (5 μL, 1.6 U/mL), and xanthine oxidase (5 μL, 10 mM), followed by the addition of 3-NBG, 2-NBP, ANBP, or AMNBP (5 μL, 5 or 10 mM). The reaction mixtures had a total volume of 0.5 mL, in which glucose/glucose oxidase were used to rapidly consume the available free oxygen, catalase was employed to remove the generated hydrogen peroxide, and xanthine/xanthine oxidase was used as the reduction agent by providing electrons [41,56]. In this way, a hypoxic enzymatic system for the reduction of the prodrugs was obtained. For the normoxic system, the composition of the reaction mixture was identical to that of the hypoxic system, except for the absence of glucose. Furthermore, nitrogen was blown into the hypoxic system for 10 s, followed by promptly sealing the tubes with airtight caps, but not into the normoxic system. The reaction mixtures were incubated at 37 °C for 0.25, 0.5, 0.75, 1, 1.5, 2, 2.5, and 3 h, followed by adding equal volume of ice-cold acetonitrile to precipitate the proteins. After standing for 15 min, the samples were centrifuged at 10,000× *g* for 10 min. Subsequently, 90 μL of the supernatant was collected and was added to 10 μL D_6_-O^6^-BG internal standard (400 nM). Finally, the reduction products were analyzed using HPLC-ESI-MS/MS.

#### 3.1.4. Determination of the Reduction Products by HPLC-ESI-MS/MS

HPLC-ESI-MS/MS was performed using a TSQ Quantum Discovery MAX triple quadrupole mass spectrometer interfaced with a SURVEYOR high-performance liquid chromatograph (Thermo Fisher Scientific, San Jose, CA, USA). A ZORBAX SB-C18 column (150 mm × 2.1 mm, 5 μm; Agilent Technologies, Palo Alto, CA, USA) was used for the separation of ABG, AMBG, and O^6^-BG by using 0.1% glacial acetic acid (solution A) and acetonitrile (solution B) as the mobile phase. The mobile phase gradient started from 95% A and was linearly reduced to 10% A over 25 min, where it was held for 5 min. The percentage of A was then increased to 95% over 3 min followed by an equilibration time of 15 min.

Mass spectrometric detection was performed in positive mode with an electrospray ionization (ESI) source. The parameters of the ESI source were set as follows: spray voltage, 3800 V; sheath gas, 30 psi; auxiliary gas, 5 psi; capillary temperature, 270 °C; tube lens offset, 113 V; and collision energy, 20 V. Positive ions were acquired in the selected reaction monitoring (SRM) mode. The signals of ABG, AMBG, O^6^-BG, and the internal standard D_6_-O^6^-BG were monitored via the transitions of *m*/*z* 257→106, 271→120, 242→91, and 248→97, respectively.

### 3.2. Theoretical Study

#### 3.2.1. Computational Methods

The geometric structures of all reactant complexes (RCs), transition states (TSs), intermediate complexes (ICs), and product complexes (PCs) were optimized using the density functional theory (DFT) method at the B3LYP/6-31+G(d,p) and M062X/6-31+G(d,p) levels [57,58,59]. All stationary points were confirmed by the same level of vibrational frequency calculation to verify that they were either TSs with only one imaginary frequency, or the energy minimum with all positive frequencies. The intrinsic reaction coordinate (IRC) was calculated at the same theoretical level of optimization to confirm that each TS connected the corresponding reactant and product through the minimized energy pathway. The zero-point correction energy (ZPE), thermal contributions to the enthalpy (Δ*H*), and Gibbs free energy (Δ*G*) were obtained from the frequency calculations. For all optimized structures, single-point calculations were performed in a chlorobenzene solution (dielectric constant *ε* = 5.6) using the conductor-like polarizable continuum model (CPCM) at the B3LYP/6-311+G(3df,2p) and M062X/6-311+G(3df,2p) levels. Cartesian coordinates for the optimized geometries and main parameters of the TSs are provided in the Appendix A. All calculations were performed using the *Gaussian 09* software package [60].

#### 3.2.2. Molecular Docking Study

X-ray crystal structure of AGT (PDB entry: 1QNT) [61] and the nitroreductase protein containing FMNH ligand (PDB entry: 4PU0) [62] were utilized as the receptors for the docking study. Hydrogen atoms were added to the protein. Gold Suite 5.2 software (Cambridge Structural Database System) was used for molecular docking, and the Gold score was chosen as the scoring function. Geometries of the ligands were optimized at the M062X/6-31+G(d,p) level using *Gaussian 09* software. The docked positions of the AGT–ligand complexes were compared with the position of human AGT (PDB entry: 1T38) in the X-ray crystal structure, which was a protein–ligand complex of AGT bound to DNA containing O^6^-methylguanine [63]. All structures were generated using *PyMOL* software (Educational version; Available online: www.pymol.org; DeLano Scientific, Scan Carlas, CA, USA).

#### 3.2.3. Molecular Dynamics Simulations

The stability of ABG, AMBG, and O^6^-BG in the active pocket of AGT were evaluated through MD simulations using the Amber 14 program package [64]. The force field parameters for protein and ligands were calculated by the AMBER FF14SB force field and the general AMBER force field (GAFF), respectively [65,66]. The geometric strain and close intermolecular contacts were relieved in the energy minimizations using the steepest descent and conjugate gradient methods. Next, each energy-minimized structure was gradually warmed from 0 to 310 K with weak constraint to the complex (5.0 kcal/mol) over 14 ps, followed by constant temperature equilibration at 310 K for 35 ps with constant volume dynamics. Subsequently, MD simulations were carried out with the periodic boundary condition in the NPT ensemble, using a non-bonded cutoff of 10 Å to truncate the VDW non-bonded interactions. Temperature (310 K) and constant pressure (1 atm) were maintained by Langevin dynamic temperature coupling with a time constant of 1.0 ps and isotropic position scaling with a relaxation time of 2.0 ps. Finally, the production step was run for 60 ns. The long-range electrostatic interactions were calculated based on the particle-mesh Ewald (PME) algorithm, and the SHAKE algorithm was applied to constrain all bonds involving hydrogen atoms [67,68].

#### 3.2.4. MM/PBSA Calculations

The MM/PBSA method was employed to evaluate the binding energies of the three AGT–ligand systems [69,70]. For each system, the binding energy (Δ*G*_binding_) was calculated for the configurations taken from a single trajectory based on the following equation:Δ*G*_binding_ = *G*_complex_ − (*G*_protein_ + *G*_ligand_) = Δ*E*_gas_ + Δ*G*_sol_ − *T*Δ*S*(1)
where the gas molecular mechanical energy (Δ*E*_gas_) was calculated as a sum of internal energies (i.e., bond, angle, and dihedral), van der Waals (*E*_vdw_), and electrostatic energies (*E*_ele_) using the SANDER module without applying a cutoff for non-bonded interactions. The solvation free energy (Δ*G*_sol_) was composed of electrostatic (Δ*G*_polar_) and non-polar (Δ*G*_non-polar_) contributions. The electrostatic contribution to the solvation free energy (Δ*G*_polar_) was determined by PB model as implemented in SANDER, applying dielectric constants of 1 and 80 to represent the solute and the exterior medium phases, respectively. The non-polar component (Δ*G*_non-polar_) was calculated using a linear function of solvent-accessible surface area (SASA) as follows: Δ*G*_non-polar_ = λ·SASA + *b*, where the corresponding parameters λ and *b* were set to 0.00542 kcal/(mol Å^2^)and 0.92 kcal/mol, respectively [71]. Given the large computational overhead and low prediction accuracy, the time-consuming conformational entropy change (−*T*Δ*S*) was not considered. The entropy term was neglected, assuming that it would be very similar for all systems.

## 4. Conclusions

In this study, four O^6^-BG derivatives were synthesized as hypoxia-activated AGT inhibitors, which were supposed to possess hypoxic reduction potential and ability to release AGT inhibitors by reduction. Using an enzymatic system to simulate the hypoxia in solid tumors, the yield of the reduction products from the four prodrugs under hypoxic conditions was observed to be significantly higher than that under normoxic conditions. The reductive activity was ANBP > AMNBP > 2-NBP > 3-NBG. It is worth noting that although the amounts of the reduction products yielded from 2-NBP and 3-NBG were lower than those from ANBP and AMNBP, their ratios of hypoxia to normoxia (C_hypoxia_/C_normoxia_ 5.55 and 6.01 for 3-NBG and 2-NBP, respectively) were higher than the other two compounds, indicating that 3-NBG and 2-NBP had superior hypoxia selectivity. In the reduction mechanism of the four prodrugs, the reduction of the nitro group to nitroso group was the rate-limiting step. The energy barriers of the rate-limiting step were 34.49 kcal/mol for 2-NBP, followed by 34.69 kcal/mol for ANBP, 35.04 kcal/mol for AMNBP and 36.88 kcal/mol for 3-NBG. The molecular docking results showed that the prodrugs had different affinities to nitroreductase (ANBP > AMNBP > 2-NBP > 3-NBG). Moreover, as shown by molecular docking and MD simulations, the reduction products had greater affinities to AGT than their parent prodrugs, which indicated that the four prodrugs would exhibit AGT-inhibitory activity after being reduced under hypoxic conditions, but not normoxic conditions. The results of in vitro experiments were generally consistent with the theoretical estimations; however, the mechanism of the hypoxia selectivity needs to be further explored as it may involve other reduction pathways under normoxic conditions. To sum up, this study not only provides new ideas for the development of novel tumor-targeted AGT inhibitors, but also demonstrated that both the reductive activity and the hypoxia selectivity should be considered in the design of hypoxia-targeted prodrugs.

## Figures and Tables

**Figure 1 ijms-20-06308-f001:**
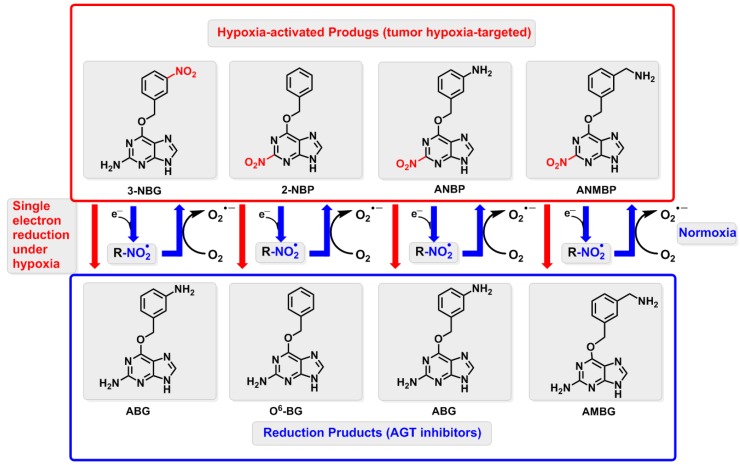
Schematic reductive process of the four prodrugs (3-NBG, 2-NBP, ANBP, and AMNBP) to release the three reduction products (O^6^-BG, ABG, and AMBG). The prodrugs are reduced to AGT inhibitors (the reduction products) via a single electron reduction under hypoxia (red line); while under normoxia, the reduction products cannot be obtained and return to the prodrugs by losing an electron to oxygen.

**Figure 2 ijms-20-06308-f002:**
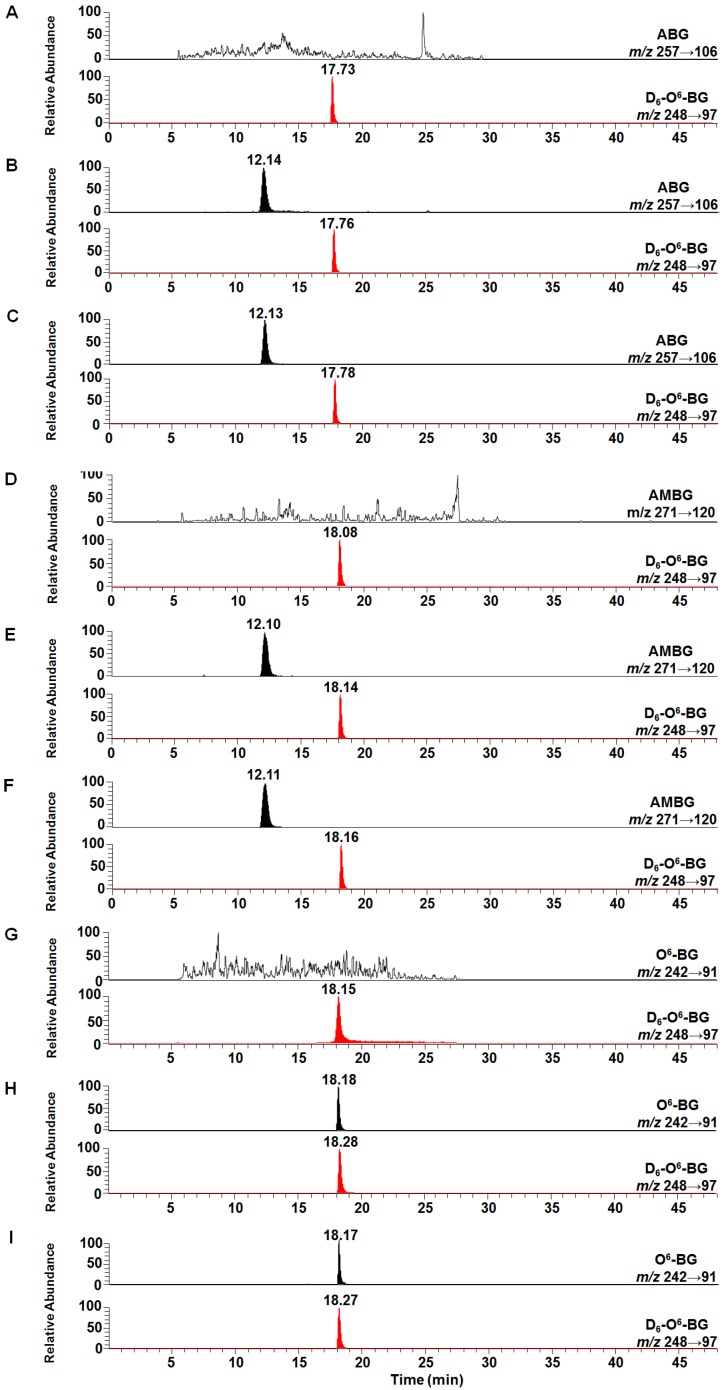
Typical selected reaction monitoring (SRM) ion chromatograms of ABG, AMBG, and O^6^-BG obtained from the four prodrugs (3-NBG, 2-NBP, ANBP, and AMNBP) and the internal standard D_6_-O^6^-BG. The prodrugs were reduced in normoxic or hypoxic condition. (**A**) Control sample without 3-NBG and ANBP. (**B**) Reduction of 3-NBG and ANBP under normoxic conditions. (**C**) Reduction of 3-NBG and ANBP under hypoxic conditions. (**D**) Control sample without AMNBP. (**E**) Reduction of AMNBP under normoxic conditions. (**F**) Reduction of AMNBP under hypoxic conditions. (**G**) Control sample without 2-NBP. (**H**) Reduction of 2-NBP under normoxic conditions. (**I**) Reduction of 2-NBP under hypoxic conditions.

**Figure 3 ijms-20-06308-f003:**
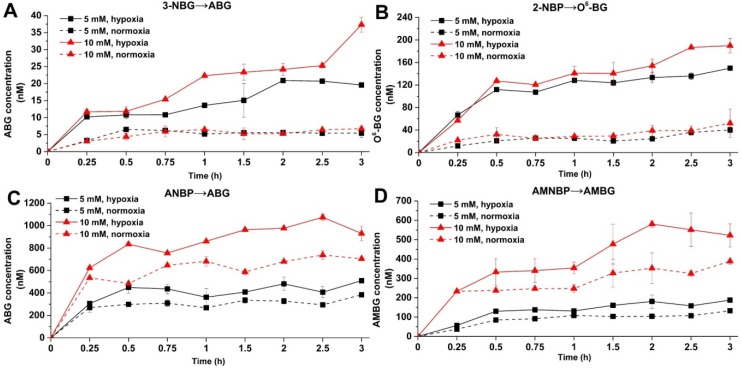
Determined concentrations of the reduction products under hypoxic (solid line) or normoxic (dash line) conditions with indicated treatment time. (**A**) ABG yielded from 3-NBG reduction. (**B**) O^6^-BG yielded from 2-NBP reduction. (**C**) ABG yielded from ANBP reduction. (**D**) AMBG yielded from AMNBP reduction. The concentrations of the prodrugs were 5 mM (black line) and 10 mM (red line).

**Figure 4 ijms-20-06308-f004:**
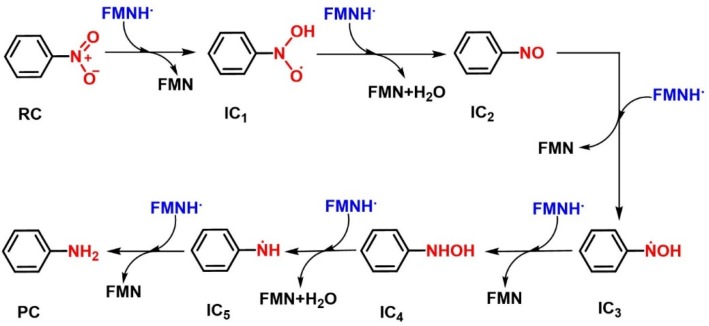
Reduction mechanism of nitrobenzene mediated by FMNH. Nitrobenzene (RC) is reduced to nitroso intermediate (IC_2_) by transferring 2e^−^/2H^+^, accompanied with elimination of a water. By transfering another 2e^−^/2H^+^, IC_2_ was converted to a hydroxylamine intermediate (IC_4_). After receiving the last 2e^−^/2H^+^, IC_4_ was converted to the aniline (PC) while eliminating a water.

**Figure 5 ijms-20-06308-f005:**
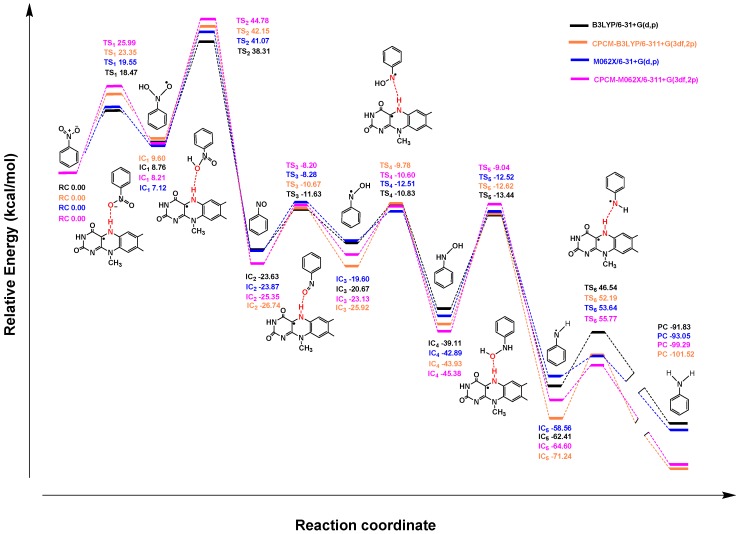
Profile of the relative energy (Δ*G*) for the reduction of nitrobenzene mediated by FMNH.

**Figure 6 ijms-20-06308-f006:**
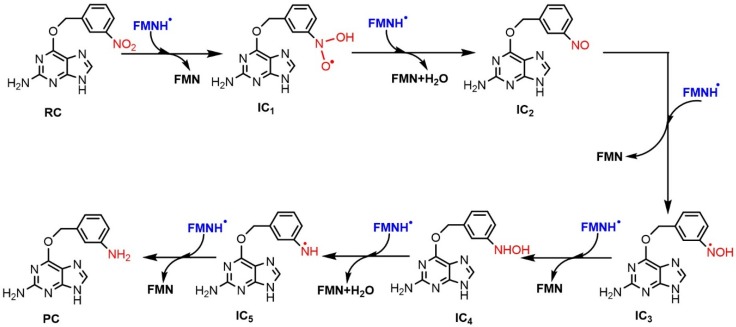
Reduction mechanism of 3-NBG mediated by FMNH. 3-NBG (RC) is reduced to nitroso intermediate (IC_2_) by transferring 2e^−^/2H^+^, accompanied with elimination of a water. By transfering another 2e^−^/2H^+^, IC_2_ was converted to a hydroxylamine intermediate (IC_4_). After receiving the last 2e^−^/2H^+^, IC_4_ was converted to the ABG (PC) while eliminating a water.

**Figure 7 ijms-20-06308-f007:**
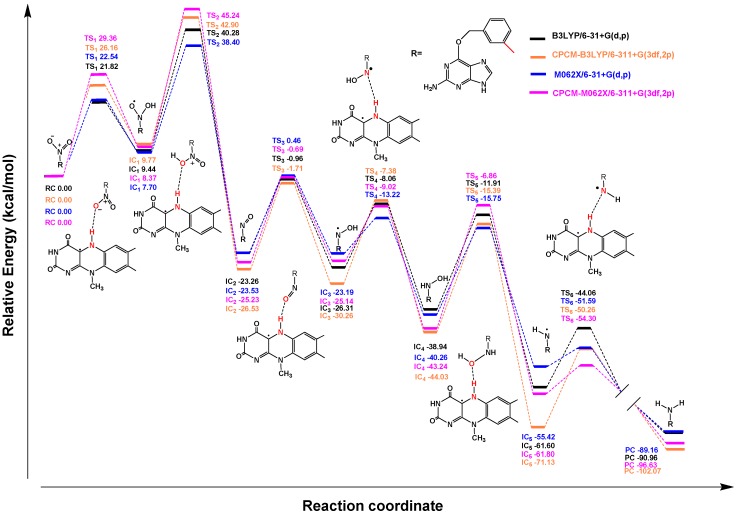
Profile of the relative energy (ΔG) for the reduction of 3-NBG mediated by FMNH.

**Figure 8 ijms-20-06308-f008:**
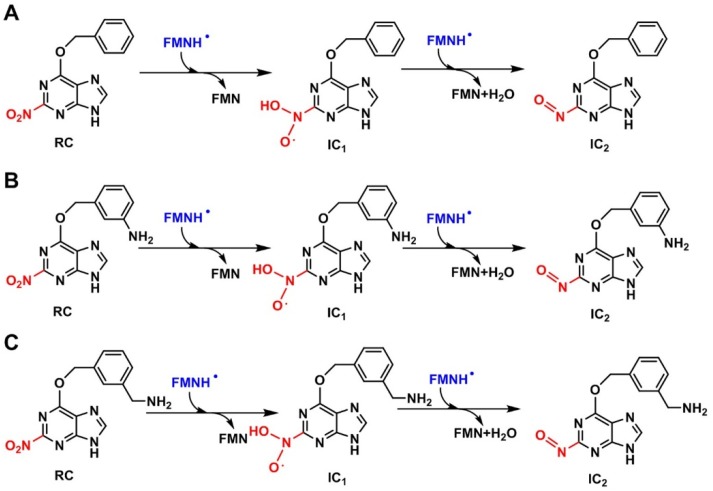
Reduction mechanisms of 2-NBP (**A**), ANBP (**B**), and AMNBP (**C**) mediated by FMNH. 2-NBP (**A**), ANBP (**B**), and AMNBP (**C**) (RC) were reduced to nitroso intermediate (IC_2_) by transferring 2e^−^/2H^+^. Meanwhile, a water molecule was eliminated.

**Figure 9 ijms-20-06308-f009:**
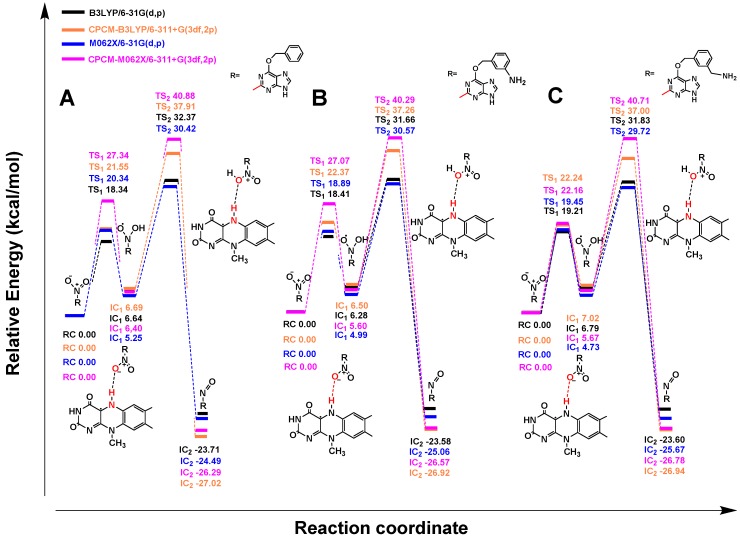
Profile of the relative energy (ΔG) for the reductions of 2-NBP (**A**), ANBP (**B**), and AMNBP (**C**) mediated by FMNH.

**Figure 10 ijms-20-06308-f010:**
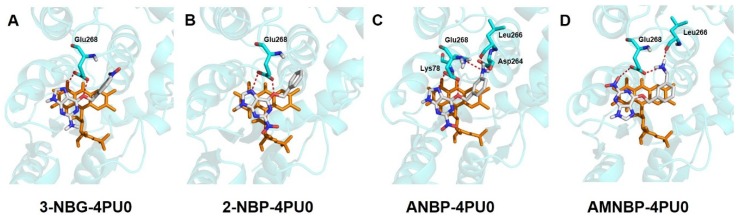
Key hydrogen bonding interactions between 3-NBG (**A**) or 2-NBP (**B**) or ANBP (**C**) or AMNBP (**D**) and the amino acid residues in nitroreductase protein (PDB entry: 4PU0). The hydrogen bonds are presented by red dotted lines and the residues forming hydrogen bonds are presented as stick models with cyan representing carbon atoms. 3-NBG, 2-NBP, ANBP, and AMNBP are presented as stick models with white representing carbon atoms. FMNH is presented as a stick model in orange. The remaining protein is displayed in the cartoon model. Nonpolar hydrogen atoms are hidden.

**Figure 11 ijms-20-06308-f011:**
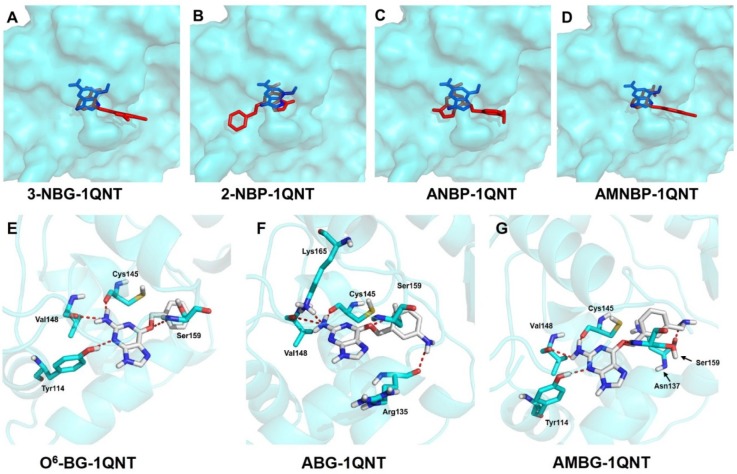
Overlap of the docked poses (in red) of the ligands (3-NBG (**A**), 2-NBP (**B**), ANBP (**C**), or AMNBP (**D**)) with the pose of O^6^-MG in the crystal structure of 1T38 (in blue). Protein is displayed as solid surface in cyan. The ligands were presented in stick models. Key hydrogen bonding interactions between O^6^-BG (**E**) or ABG (**F**) or AMBG (**G**) and the amino acid residues in the active pocket of AGT are represented by red dotted lines, and the residues are presented as a stick model with cyan representing carbon atoms. O^6^-BG, ABG, and AMBG are presented as stick models with white representing carbon atoms. The remaining protein is displayed in a cartoon model. Nonpolar hydrogen atoms are hidden.

**Figure 12 ijms-20-06308-f012:**
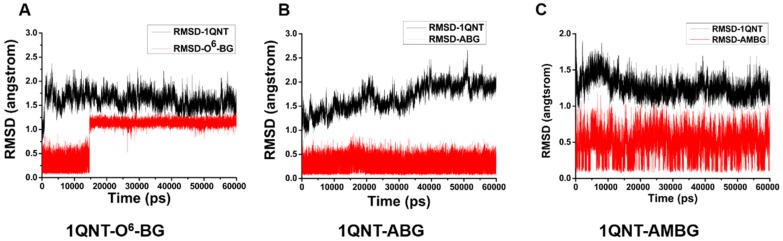
The time dependence of root mean square deviation (RMSD) of inhibitors (red line) and AGT (black line) for AGT–ligand systems with O^6^-BG (**A**), ABG (**B**), and AMBG (**C**).

**Table 1 ijms-20-06308-t001:** Energy barriers (ΔG) of the rate-limiting step (TS_2_) in the reduction of the four prodrugs mediated by FMNH under hypoxia ^a^.

Theoretical Levels	3-NBG	2-NBP	ANBP	AMNBP
B3LYP/6-31+G(d,p)	30.83	25.73	25.37	25.04
CPCM-B3LYP/6-311+G(3df,2p)	33.13	30.95	30.76	29.98
M062X/6-31+G(d,p)	30.71	25.18	25.57	24.99
CPCM-M062X/6-311+G(3df,2p)	36.88	34.49	34.69	35.04

^a^ All energies are in kcal/mol.

**Table 2 ijms-20-06308-t002:** Docking results of four prodrugs with nitroreductase protein.

Molecule	Fitness Score	Hydrogen Bonds		
		Amount	Length/Å	
3-NBG	49.5377	2	O^6^-Glu268	2.4
			N7-Glu268	2.9
2-NBP	49.9341	2	O^6^-Glu268	2.1
			N7-Glu268	2.8
ANBP	53.4563	5	O^6^-Glu268	2.3
			N7-Glu268	2.8
			N-Leu266	2.6
			N-Asp264	2.4
			N-Lys78	3.1
AMNBP	53.2185	3	O-Glu268	2.9
			N-Glu268	2.8
			N-Leu266	2.8

**Table 3 ijms-20-06308-t003:** Docking results of the four prodrugs and their reduction products with AGT.

Compounds	Fitness Score	Orientation ^1^	Hydrogen Bonds		
			Amount	Length/Å	
3-NBG	47.0434	No	/	/	
2-NBP	49.7631	No	/	/	
ANBP	49.4182	No	/	/	
AMNBP	50.8505	No	/	/	
**O^6^-BG**	54.8831	Yes	4	O^6^–Ser159	3.2
				N^2^–Val148	3.0
				N^2^–Cys145	2.4
				N^4^–Tyr114	2.9
**ABG**	53.3393	Yes	5	N^2^–Lys165	3.0
				O^6^–Ser159	2.9
				N^2^–Val148	3.0
				N^2^–Cys145	2.5
				N–Arg135	3.0
**AMBG**	53.9127	Yes	5	O^6^–Ser159	2.0
				N^2^–Val148	2.4
				N^2^–Cys145	1.7
				N–Asn137	1.7
				N^3^–Tyr114	2.1

^1^ “Yes” represents the pose of the compounds in the active pocket of AGT protein being similar to the pose of the ligand in the crystal structure of AGT (PDB entry: 1T38) and agreeing with the repairing mechanism of AGT; “No” represents the opposite of “Yes”.

**Table 4 ijms-20-06308-t004:** Energy terms of MM/PBSA results for three AGT–ligand systems.

Free Energy (Kcal/mol)	O^6^-BG	ABG	AMBG
ΔE_ele_	−20.72 ± 1.35	−37.54 ± 1.74	−34.92 ± 1.74
ΔE_vdw_	−38.57 ± 1.35	−40.55 ± 1.89	−39.19 ± 2.67
ΔE_gas_ ^a^	−59.29 ± 1.60	−78.09 ± 3.62	−74.12 ± 2.64
ΔG_nonpolar_	−4.35 ± 0.06	−4.43 ± 0.04	−5.05 ± 0.20
ΔG_polar_	30.68 ± 1.16	46.90 ± 2.53	43.71 ± 1.74
ΔG_sol_ ^b^	26.33 ± 1.14	42.47 ± 2.49	38.66 ± 1.74
ΔG_ele_ ^c^	9.96 ± 1.97	9.36 ± 0.79	8.79 ± 1.76
ΔG_binding_ ^d^	−32.96 ± 1.75	−35.63 ± 1.13	−35.45 ± 3.30
ΔΔG_binding_ ^d^	0	−2.67	−2.49

^a^ ∆E_gas_ = ∆E_ele_ + ∆E_vdw_; ^b^ ∆G_sol_ = ∆G_polar_ + ∆G_nonpolar_; ^c^ ∆G_ele_ = ∆E_ele_ + ∆G_polar_; ^d^ ∆G_binding_ = ∆E_ele_ + ∆E_vdw_ + ∆G_sol_.

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
