# Peer review of "Reductive Activity and Mechanism of Hypoxia- Targeted AGT Inhibitors: An Experimental and Theoretical Investigation"

_ijms, 2019, doi:10.3390/ijms20246308_

Round 1
Reviewer 1 Report
In my personal perspective the revised work entitled "Reductive Activity and Mechanism of Hypoxia-Targeted AGT Inhibitors: An Experimental and Theoretical Investigation" is well written, interesting and describes in a constructive manner the new prodrugs. I believe that the new additions (ie MD) and the full experimental into the main article consists of a major improvement in the document. Therefore, I highly recommend the present work in order to be published in IJMS.
Author Response
We are very grateful to the reviewer for giving us positive comments and recommending our work to be published in IJMS.
Reviewer 2 Report
The article entitled: Reductive Activity and Mechanism of Hypoxia Targeted AGT Inhibitors: An Experimental and Theoretical Investigation has taken in to account the significant problem for the anticancer therapy. The main target of new synthetized compounds was
O6-alkylguanine-DNA alkyltransferase, protein response for the tumour cell resistance to DNA alkylating agents. Alkylating drugs are the important force in the chemo or chemo/radiotherapy strategy. In my opinion the presented results have given the good background of the hypoxia-activated mechanism of nitro-substituted proinhibitors of AGT as well as may be useful for further design of novel anticancer molecules.
The experimental part of the manuscript is readable however I have some comments:
from the physiological point of view the MD should be performed in temperature 310K instead of 300K the geometry and reaction path should be calculated in water as condensed phase not chlorobenzene the IRC strategy should be performed. from the organic synthetic site I did not find any information about 13C NMR analysis/spectra. I recommended to discuss: why the Minnesota functional and B3LYP were used and what is difference between them, maybe some references?In conclusion the article is well written and I hope that authors provide my comments which make the article suitable for publication.
Author Response
Manuscript ID: IJMS-645293
Title: Reductive Activity and Mechanism of Hypoxia-Targeted AGT Inhibitors: An Experimental and Theoretical Investigation
Dear Editor-in-Chief:
Thank you very much for your work on our manuscript and conveying the reviewers’ comments to us. We appreciate the reviewers for their instructive comments. The advices from the reviewers are helpful for revising our manuscript. According to the reviewers’ comments, we revised the manuscript carefully. The modifications and the explanations for the concerns from the reviewers are listed point-by-point as follows.
Response to the comment of Reviewer #2
The article entitled: Reductive Activity and Mechanism of Hypoxia Targeted AGT Inhibitors: An Experimental and Theoretical Investigation has taken in to account the significant problem for the anticancer therapy. The main target of new synthetized compounds was O6-alkylguanine-DNA alkyltransferase, protein response for the tumour cell resistance to DNA alkylating agents. Alkylating drugs are the important force in the chemo or chemo/radiotherapy strategy. In my opinion the presented results have given the good background of the hypoxia-activated mechanism of nitro-substituted proinhibitors of AGT as well as may be useful for further design of novel anticancer molecules. In conclusion the article is well written and I hope that authors provide my comments which make the article suitable for publication. The experimental part of the manuscript is readable however I have some comments:
1) from the physiological point of view the MD should be performed in temperature 310K instead of 300K the geometry and reaction path should be calculated in water as condensed phase not chlorobenzene the IRC strategy should be performed.
Author response: We are so appreciated to the reviewer for giving us positive comments. The concerns from the reviewer were answered point-by-point and several modifications were performed according to the reviewer’s suggestion.
For the reviewer’s concern about the temperature in MD, we carefully checked the parameters in the MD calculations and found that the temperature 300K was written by mistake. It should be 310K. We thank the reviewer for giving us an important reminder, and therefore 310K has been modified to 300K in the revision (line 404 to line 408). We understand the reviewer’s concern that reaction pathways were mostly calculated in water as condensed phase. However, the hypoxic reduction of the substrates synthesized in this work with FMNH occurs in the active pocket of nitroreductase protein, but not in water solution; and the dielectric constant of protein (ε = 2~40) is much smaller than that of water (ε = 78). Therefore, according to previous studies,[Ji L., et al. J. Phys. Chem. B 2012, 116, 903-712. Schutz C.N., et al. Proteins 2001, 44, 400-417.] the reaction pathways of the four substrates (3-NBG, 2-NBP, ANBP and ANMBP) reduced by FMNH were calculated in chlorobenzene (ε = 5.6) as the condensed phase. As mentioned by the reviewer, we did perform IRC calculations for all transition states (TS) at the B3LYP/6-31+G(d,p) and M062X/6-31+G(d,p) theoretical levels, which was omitted to describe in the manuscript. Therefore, in the revision (in line 380-383), we added “The intrinsic reaction coordinate (IRC) was calculated at the same theoretical level of optimization to confirm that each TS connects the corresponding reactant and product through the minimized energy pathway.”
2) From the organic synthetic site I did not find any information about 13C NMR analysis/spectra.
Author response: We understand the reviewers concern about the 13C-NMR spectra. However, considering the data of 1H-NMR, IR and high-resolution MS were sufficient to confirm the correct molecular structure of the synthesized compounds, we didn’t perform 13C-NMR analysis in this work so as to save the compounds for the following hypoxic reduction experiments. As described in the supplementary materials, the data of the 1H-NMR, IR and high-resolution MS analysis demonstrated that all compounds had correct chemical structures.
3) I recommended to discuss: why the Minnesota functional and B3LYP were used and what is difference between them, maybe some references?
Author response: There are two reasons for the application of B3LYP and Minnesota functional in this study. Firstly, as a classic DFT method, B3LYP is widely used in various theoretical calculations. M062X as another popular DFT method, has good advantages in the calculations of transition states and structure optimizations. Therefore, we used both B3LYP and M062X to explore the reaction mechanisms and to examine whether there is any difference between the results obtained from the two methods. Secondly, M062X and B3LYP were simultaneously used in many investigations on the mechanisms involving radical-reaction systems, in which little difference was observed between the results obtained from the two methods as those observed in our study.[Anbazhakan K., et al. Struct. Chem. 2019, 30, 167-173. Diaz, M.G., et al. Struct. Chem. 2019, 30, 237-245. Zhao L.J., et al. Int. J. Quantum Chem. 2013, 113, 1299-1306] This means that both M062X and B3LYP are suitable for the mechanisms of the reductive mechanism of the four hypoxia-targeted AGT inhibitors in this study. As suggested by the reviewer, the following literatures were added in the manuscript as references 57-59.
References:
[57] Lee, C.T.; Yang, W.T.; Parr, R.G. Development of the Colic-Salvetti correlation-energy formula into a functional of the electron density. Phys. Rev. B 1987, 37, 785-789.
[58] Becke, A.D. Density-functional thermochemistry. III. The role of exact exchange. J. Chem. Phys. 1992, 98, 5648-5652.
[59] Zhao, Y; Truhlar, D.G. The M06 suite of density functionals for main group thermochemistry, thermochemical kinetics, noncovalent interactions, excited states, and transition elements: two new functionals and systematic testing of four M06-class functionals and 12 other functionals. Theor. Chem. Acc. 2008, 120, 215-241.
We are very appreciated for your work on our manuscript. If you have any question about our manuscript, please do not hesitate to contact with me. We are looking forward to receiving information from you. Best regards for you,
This manuscript is a resubmission of an earlier submission. The following is a list of the peer review reports and author responses from that submission.
Round 1
Reviewer 1 Report
The present review describes the Reductive Activity and Mechanism of Hypoxia-Targeted AGT Inhibitors: An Experimental and Theoretical Investigation. The overall work is interesting, describing in a constructive manner the new prodrugs. It is well written, and in my personal opinion, could be published likewise in Molecules. Nevertheless, the authors should check some modifications, especially in the supplementary section, as mentioned bellow:
Line 263: The authors claim that This suggested the loss of AGT inhibitory activity of the prodrugs before reduction, which meant that the prodrug would not inhibit AGT activity under normoxic condition.
In to my opinion they have to rephrase. What did the inhibitors lose? Probably they mean that the prodrugs are inactive against AGT under normoxic conditions since only the reductive form inhibits AGT activity
Line 272: The hydrogen bond between ABG and Tyr114 was disappeared, What do they mean disappeared?
Line 274: bond
Line 276: bond
Line 309: is
Line 325: supernatant was removed and was added with 10 μL D6-O6-BG ???
Line 369: Using an enzymatic system for simulating the hypoxia in solid tumors, the yielding of the reduction products from the four prodrugs under hypoxic condition was observed to be significantly higher than that under normoxic condition, with the reductive activity as ANBP. AMNBP > 2-NBP > 3-NBG. Yield instead of yielding. In general, this sentence is confusing. Additionally, is preferable, for your conclusion to change the tense: It is worth noting or worth to note…
In the reduction mechanism of the four prodrugs, the reduction of the nitro group to nitroso group was the rate-limiting step: is the rate etc
Regarding the experimental for the synthesis of the compounds on the supplementary data, the authors have to optimize the document. These data are crucial for the right presentation of their work. They have to work a lot with the experimental:
Some examples but not all:
Compound 2: the right name is 1-methyl-1-(9H-purin-6-yl) pyrrolidin-1-ium chloride
For all the compound H have to be in italics
As for the experimental they have to consider something like:
To a solution of compound 1 (909 mg, 5.9 mmol) in DMF (40 mL) 1-methylpyrrolidine (1.4 mL) was added and the mixture is stirred at room temperature for 18 h. After completion of the reaction, the obtained crystals were filtrated and were recrystallized by ether, which gave compound 2 as a white solid (1041 mg, 5.1 mmol, yield 86%).
This goes for all the experimental.
Another paradigm goes for the next compound 3: what do they mean with deionized water containing (440 μL) glacial acetic acid and excess potassium tert-butoxide.
For compound 6: add (852 mg, 2.8 mmol) tetrabutylammonium nitrate, drop (0.5 mL) dichloromethane containing (366 μL, 1.2 mmol) of trifluoroacetic anhydride for 1-2 min and r
The name of compound 10 is (6-((3-(2,2,2-trifluoroacetamido)benzyl)oxy)-9H-purin-9-yl)methyl pivalate
And the right structure is
If the protective group is the chloromethylpivalate.
The same goes for compounds 11, 15 and 16.
And finally one question: The authors have in vitro the reduction of their prodrugs but for the inhibitory activity they have only in silico results. Will they evaluate in vitro this activity?
Reviewer 2 Report
In the present manuscript the authors provide the synthesis and exploration of reductive activity and mechanism of hypoxia of four potential AGT inhibitors. To accomplish this goal, the authors use a combination of experimental assays and molecular modeling techniques (quantum chemistry calculations and molecular docking). The manuscript and the results provided are interesting, however, I think that the activity of compounds need to be proved experimentally and/or with more accurate computational methods to merit publication in International Journal of Molecular Sciences.
Major comments:
The activity of the synthesized compounds is not tested. Molecular docking scores and poses are not reliable to assess the activity of an inhibitor. Either experimental testing and/or more elaborated computational protocols should be used to assess the activity of the synthesized compounds.
The activity of the four synthesized compounds should be experimentally evaluated. The introduced modifications on the O6-BG scaffold can decrease the affinity of the compounds towards the active site of both nitroreductase and AGT proteins. Molecular docking can sometimes not capture the role of subtle changes. In addition, the small differences observed in the Fitness Scores of the four compounds cannot be used to make strong statements (direct comparison with the experimental trends for example). A more robust approach should be followed. Molecular dynamics simulations should be performed to assess the stability of the compound in the active site. From MD simulations the stability of the predicted binding pose will be confirmed. From the MD simulations, MM/P(G)B-SA binding affinities can be easily computed that will provide more reliable results than docking scores. Therefore, I recommend performing MD simulations followed by some binding affinity calculations to prove the statements made based on molecular docking.Regarding the quantum chemistry calculations, the models used to study the reduction mechanism are simple in order to be computationally affordable. The results obtained assigned similar rate-limiting step barriers for all compounds and this does not allow to differentiate them as in experimental results. Some features of the active site of the nitroreductase enzyme may be key to differentiate between the analyzed compounds. Bulkier substituents in the phenyl moiety of the prodrugs may be key to properly orient the molecule in the active site for reduction and this cannot be captured in the simple models suggested (Figure 10 apparently indicates that the amino group of the prodrug is stabilized by Glu268, Leu266, …). The authors should clarify this simplification and why not using small cluster models (theozyme, extracted from docking poses for example) or QM/MM (based on docking poses of section 2.3). Maybe the inclusion of some amino acid residues will providea a clear trend between the different compounds.
Minor comments:
The four synthesized compounds are O6-BG derivatives. However, it is not specified the criteria used to select ABG and AMBG compounds. Are these compounds active in previous screenings?
Additional minor details on quantum chemistry calculations.
A justification of the four levels of theory used to perform the QM calculations should be provided. The description of the results is focused on CPCM-M062X/6-311+G(3df,2p) but it is also the method that present smaller differences in the energy barriers of the rate limiting step. I wonder if the authors have checked the spin density on the transition state structures to assess the radical character of the transition states depicted in Figures 7 and 9.The molecular basis of the inhibition process should be discussed in more detail. The analysis of interactions between the drug and the receptor based on docking calculations is scarce and limited to hydrogen bonds (are there other important interactions? π-stacking, hydrophobic, ... MD simulations can help on this analysis to provide information on the stability of the mentioned interactions along time.
